# Protective Effects of the Chalcone-Based Derivative AN07 on Inflammation-Associated Myotube Atrophy Induced by Lipopolysaccharide

**DOI:** 10.3390/ijms232112929

**Published:** 2022-10-26

**Authors:** Wei-Yu Fang, Chih-Lung Lin, Wan-Hsuan Chang, Chih-Hsiang Chang, Yun-Cian Huang, Yi-Hong Tsai, Fang-Rong Chang, Yi-Ching Lo

**Affiliations:** 1Department of Pharmacology, School of Medicine, College of Medicine, Kaohsiung Medical University, Kaohsiung 80708, Taiwan; 2Department of Neurosurgery, School of Medicine, College of Medicine, Kaohsiung Medical University, Kaohsiung 80708, Taiwan; 3Division of Neurosurgery, Department of Surgery, Kaohsiung Medical University Hospital, Kaohsiung 80708, Taiwan; 4Graduate Institute of Medicine, College of Medicine, Kaohsiung Medical University, Kaohsiung 80708, Taiwan; 5Graduate Institute of Natural Products, College of Pharmacy, Kaohsiung Medical University, Kaohsiung 80708, Taiwan; 6Department of Medical Research, Kaohsiung Medical University Hospital, Kaohsiung 80708, Taiwan

**Keywords:** chalcone, inflammation, myotube, mitochondrial oxygen consumption, protein synthesis/degradation

## Abstract

Inflammation is a major cause of skeletal muscle atrophy in various diseases. 2-Hydroxy-4′-methoxychalcone (AN07) is a chalcone-based peroxisome-proliferator-activated receptor gamma (PPARγ) agonist with various effects, such as antiatherosclerosis, anti-inflammation, antioxidative stress, and neuroprotection. In this study, we examined the effects of AN07 on protein homeostasis pathway and mitochondrial function in inflammation-associated myotube atrophy induced by lipopolysaccharides (LPS). We found that AN07 significantly attenuated NF-κB activation, inflammatory factors (TNF-α, IL-1β, COX-2, and PGE2), Nox4 expression, and reactive oxygen species levels in LPS-treated C2C12 myotubes. Moreover, AN07 increased SOD2 expression and improved mitochondrial function, including mitochondrial membrane potential and mitochondrial oxygen consumption rate. We also demonstrated that AN07 attenuated LPS-induced reduction of myotube diameter, MyHC expression, and IGF-1/IGF-1R/p-Akt-mediated protein synthesis signaling. Additionally, AN07 downregulated LPS-induced autophagy–lysosomal protein degradation molecules (LC3-II/LC3-I and degraded p62) and ubiquitin–proteasome protein degradation molecules (n-FoxO1a/MuRF1/atrogin-1). However, the regulatory effects of AN07 on protein synthesis and degradation signaling were inhibited by the IGF-1R inhibitor AG1024 and the PI3K inhibitor wortmannin. In addition, the PPARγ antagonist GW9662 attenuated the effects of AN07 against LPS-induced inflammation, oxidation, and protein catabolism. In conclusion, our findings suggest that AN07 possesses protective effects on inflammation-induced myotube atrophy and mitochondrial dysfunction.

## 1. Introduction

Skeletal muscle atrophy is caused by various common diseases or conditions, such as disuse from illness or injury, sepsis, malnutrition, metabolic syndrome, cachexia, aging, and coronavirus disease 2019 (COVID-19) infection [1,2], which leads to a decline in physical activity performance and results in increased morbidity, mortality, and health care costs [3,4]. Due to the multifactorial pathogenesis of skeletal muscle atrophy, current medical treatments for improving muscle atrophy are still limited. Inflammation, a key causative factor in many diseases, is one of the critical factors leading to skeletal muscle atrophy [5,6]. Therefore, the development of anti-inflammatory agents with antiatrophic effects could contribute to inflammation-induced muscle atrophy.

Imbalance in protein synthesis and degradation pathways is strongly associated with inflammation-induced muscle wasting. It is well known that insulin-like growth factor 1 (IGF-1)/type 1 IGF receptor (IGF-1R) and its downstream phosphoinositide 3-kinase (PI3K)/Akt signaling are involved in protein synthesis of skeletal muscle [7]. Lipopolysaccharide (LPS) is commonly used to induce myotube atrophy via increase in inflammatory mediators [8,9,10] and downregulation of IGF-1/Akt/mTOR signaling [11,12]. Moreover, LPS induces protein degradation in the skeletal muscle via activation of the ubiquitin–proteasome system (UPS) and the autophagy–lysosome system (ALS) [13,14]. In atrophic skeletal muscle, two muscle-specific E3 ubiquitin ligases, namely, muscle atrophy F-box (atrogin-1/MAFbx) and muscle ring finger 1 (MuRF1), are highly upregulated by Forkhead box protein O1A (FoxO1a) transcription factors [15,16]. LPS also activates the atrogin-1/MuRF1/FoxO1a pathway, leading to skeletal muscle atrophy [17,18]. In addition, FoxO1a activates another proteolytic system, ALS [15,19]. When autophagy is activated, the cytosolic-microtubule-associated protein 1-light chain 3-I (LC3-I) conjugates with phosphatidylethanolamine to form LC3-II, which is subsequently recruited to the membranes of autophagosome to initiate the formation and lengthening of autophagosome [20]. In LPS-treated C2C12 myotubes and mouse skeletal muscle, the upregulation of autophagy signaling has been observed [8,21]. Oxidative stress caused by the imbalance between reactive oxygen species (ROS) production and antioxidant defense system is an important contributor to inflammation-induced muscle wasting [22]. NADPH oxidase 4 (Nox4), an enzyme that catalyzes the transfer of electrons from NADPH to oxygen and then generates superoxide [23], has been identified as a primary source of LPS-induced ROS production in many types of cells [24,25]. In addition, LPS-induced ROS increases NF-κB-dependent inflammatory mediators, which further increases ROS production by altering the redox homeostasis system [26]. Furthermore, the increase in ROS production leads to loss of membrane potential and damage of mitochondrial respiratory chain [27,28], which results in a decrease in the mitochondrial oxygen consumption rate [29]. On the other hand, LPS decreases superoxide dismutase (SOD), an antioxidant enzyme that catalyzes the reduction of superoxide anions to hydrogen peroxide [30].

Chalcones (1,3-diphenyl-2-propen-1-ones), a subclass of open-chain flavonoids, are one of the most privileged scaffolds in medicinal chemistry that can be synthesized in the laboratory and converted into several therapeutically active heterocyclic scaffolds [31]. Based on their simple chemistry and convenient synthesis, many chalcone derivatives have been prepared and found to exert a broad spectrum of activities, such as antidiabetic, anti-inflammatory, antioxidant, antitumor, anti-infective, and anti-Alzheimer’s disease, in different disease models [32,33]. However, to date, the role and mechanism of chalcones on skeletal muscle remains unclear. 2-Hydroxy-4′-methoxychalcone (AN07, Figure 1), a synthetic chalcone, possesses anti-inflammatory and antiatherosclerotic activities in human aortic smooth muscle cells via activation of peroxisome-proliferator-activated receptor-γ (PPARγ) [34]. PPARγ is a ligand-activated transcription factor belonging to the PPAR nuclear receptor superfamily and controls the expression of genes involved in various biological functions, including energy regulation, lipid metabolism, insulin sensitivity, oxidative stress, and inflammatory response [35]. It has been shown that upregulation of PPARγ inhibits TNF-α-mediated NF-κB activation and attenuates TNF-α-impaired C2C12 differentiation [36]. Moreover, activation of PPARγ induces gene expression of glutathione peroxidase 3, leading to a decrease in ROS accumulation and ROS-mediated insulin resistance in human skeletal muscle cells [37]. Recently, PPARγ has been found to increase IGF-1-mediated protein synthesis through the inhibition of miR-29b [38]. According to the above findings, PPARγ agonist might serve as a potential pharmacological agent to protect inflammation-induced myotube atrophy. Our previous study also showed the anti-inflammatory, antioxidant, and neuroprotective properties of AN07 in LPS-stimulated RAW264.7 macrophages and methylglyoxal-treated SH-SY5Y neural cells [39]. In this study, we further examined the protective effects of AN07 on LPS-induced myotube atrophy and the responsible molecular mechanisms. The present study demonstrates that AN07 not only attenuates LPS-induced myotube atrophy through its anti-inflammatory and antioxidant properties but also promotes protein synthesis and reduces protein degradation, suggesting that AN07 has the novel potential to improve inflammation-induced myotube atrophy.

## 2. Results

### 2.1. AN07 Attenuated LPS-Induced C2C12 Myotube Atrophy

First, we examined the effects of AN07 (0.01–1 μM) on cell viability and cytotoxicity of C2C12 myotubes by MTT and LDH tests, respectively. As shown in Figure 2A,B, AN07 (0.01–1 μM) did not affect the cell viability and cytotoxicity of myotubes treated with AN07 for 24 h. Then, we investigated the effect of AN07 on LPS-treated myotubes. As shown in Figure 2C, LPS induced significant morphological atrophy of myotubes. Moreover, LPS did not affect the myotube number (Figure 2D) but reduced myotube diameter (Figure 2E). However, AN07 attenuated LPS-induced reduction of myotube diameter (Figure 2E). LPS also downregulated MyHC protein expression of myotubes, whose effect was attenuated by AN07 treatment (Figure 2F).

### 2.2. AN07 Attenuated LPS-Induced Oxidative Stress in C2C12 Myotubes

To investigate the effects of AN07 on LPS-induced oxidative stress in C2C12 myotubes, the intracellular ROS level was measured by H_2_DCF-DA fluorescence staining. As shown in Figure 3A, LPS (100 ng/mL) increased ROS production compared to the vehicle group, but AN07 (0.5–1 μM) attenuated LPS-induced ROS production. In addition, AN07 attenuated LPS-induced upregulation of Nox4 expression (Figure 3B) and downregulation of SOD2 expression, an antioxidant protein (Figure 3C), in C2C12 myotubes. We further evaluated the effects of AN07 on mitochondrial membrane potential (ΔΨm) of LPS-treated C2C12 myotubes by measuring the ratio of JC-1 red and green fluorescence (Figure 3D). The results indicated that LPS decreased ΔΨm of C2C12 myotubes, whose effect was attenuated by AN07 (0.1–1 μM) treatment (Figure 3E).

### 2.3. AN07 Improved Mitochondrial Respiration Function in LPS-Treated C2C12 Myotubes

To investigate the effect of AN07 on mitochondrial respiration function, the oxygen consumption rate (OCR), a metabolic parameter representing mitochondrial respiration levels, was measured by the Seahorse XFp analyzer. As shown in Figure 4A, following the measurement of OCR for basal respiration, oligomycin (ATP synthase inhibitor) was added to measure the OCR linked to ATP production. Subsequently, FCCP (an uncoupling agent) was added to evaluate the maximal respiration OCR. Finally, rotenone (complex I inhibitor) and antimycin A (complex III inhibitor) were mixed and added to block mitochondrial electron transport chain reaction. In response to LPS, cellular basal OCR (Figure 4B), ATP production OCR (Figure 4C), and maximal respiration OCR (Figure 4D) were remarkably reduced in C2C12 myotubes. However, the addition of AN07 restored LPS-induced reduction of OCR parameters, including basal respiration, ATP production, and maximal respiration OCR (Figure 4B–D).

### 2.4. AN07 Attenuated LPS-Induced Inflammatory Signaling in C2C12 Myotubes

Next, we examined the effects of AN07 in LPS-induced inflammatory signaling in myotubes. As shown in Figure 5A, LPS significantly increased the nuclear NF-κBp65 levels and decreased the cytosol levels of NF-κBp65, accompanied by an increase in IκBα phosphorylation in the C2C12 myotubes (Figure 5B). Moreover, LPS upregulated the downstream effectors of NF-κB, such as TNF-α (Figure 5C), IL-1β (Figure 5D), and COX-2 (Figure 5E). However, AN07 attenuated the nuclear expression of n-NF-κB (Figure 5A) and expression of p-IκBα (Figure 5B), TNF-α (Figure 5C), and IL-1β (Figure 5D) induced by LPS. Moreover, AN07 decreased LPS-induced COX-2 expression (Figure 5E) and PGE2 production (Figure 5F).

### 2.5. AN07 Activated IGF/IGF-1R-Related Protein Synthesis Pathway in LPS-Treated C2C12 Myotubes

To further understand the molecular mechanisms responsible for the protective effects of AN07 on LPS-induced myotube atrophy, the involvement of the protein synthesis pathway IGF-1/IGF-1R was investigated. As shown in Figure 6A, LPS induced downregulation of IGF-1R protein expression, but AN07 upregulated IGF-1R expression in LPS-treated myotubes. AN07 also enhanced the IGF-1R downstream signaling molecules IRS-1 and p-Akt (Figure 6B,C). Additionally, LPS or AN07 treatment for 24 h did not affect p-mTOR (S2448) (Figure 6D) and p-S6 (S235/236) protein levels (Figure 6E). Therefore, we further measured the p-mTOR (S2448) and p-S6 (S235/236) protein levels after LPS or AN07 treatment for 12 h. The results showed that LPS decreased both p-mTOR (S2448) (Figure 6F) and p-S6 (S235/236) (Figure 6G) protein levels, whose effects were attenuated by AN07. In addition, AN07 attenuated LPS-induced reduction of IGF-1 mRNA expression (Figure 6H).

### 2.6. AN07 Suppressed LPS-Induced Protein Degradation Pathway in C2C12 Myotubes

As shown in Figure 7A, LPS increased the nuclear translocation of FoxO1a, whose effect was attenuated by AN07. In addition, LPS enhanced the UPS pathway by upregulation of MuRF1 (Figure 7B) and atrogin-1 (Figure 7C). However, AN07 attenuated LPS-induced protein expression of MuRF1 (Figure 7B) and atrogin-1 (Figure 7C). Moreover, AN07 attenuated LPS-induced ALS pathway signaling by downregulation of LC3B-II protein expression (Figure 7D). Furthermore, AN07 attenuated LPS-induced downregulation of p62/SQSTM1, an autophagic marker (Figure 7E).

### 2.7. Effects of PPARγ, IGF-1R, and PI3K/Akt Antagonists on AN07-Induced Protective Effects in LPS-Treated C2C12 Myotubes

To examine the involvement of PPARγ in the protective effects of AN07 on LPS-induced inflammatory/oxidative response and atrophic-related molecules, the selective PPARγ antagonist GW9662 was used. The results showed that GW9662 (10 μM) significantly attenuated the inhibiting effects of AN07 (1 μM) on the protein expression of COX-2 (Figure 8A) and Nox4 (Figure 8B) in LPS-treated myotubes. In addition, GW9662 attenuated the enhancing effect of AN07 on the protein expression of MyHC (Figure 8C) and IGF-1R (Figure 8D) and the inhibitory effects of AN07 on MuRF1 (Figure 8E) in LPS-treated myotubes. Then, we further investigated the involvement of IGF-1R/Akt signaling in the antiatrophic mechanisms of AN07 using the IGF-1R antagonist AG1024 (10 nM) and the PI3K/Akt inhibitor wortmannin (100 nM). The results indicated that AG1024 and wortmannin significantly attenuated the promoting effects of AN07 (1 μM) on myotubes diameter (Figure 9A,B) and MyHC protein expression (Figure 9C). In addition, AG1024 and wortmannin reversed the inhibitory effects of AN07 on MuRF-1 expression (Figure 9D). However, AG1024 and wortmannin did not affect the inhibitory effects of AN07 on COX-2 (Figure 9E) and Nox4 (Figure 9F) expression in LPS-treated C2C12 myotubes.

## 3. Discussion

In this study, we demonstrated that AN07 protected myotubes against LPS-induced atrophy by regulating protein synthesis and degradation signaling, attenuating inflammatory and oxidative stress signaling, and improving mitochondrial function. The balance between Noxs-induced ROS production and SODs-induced ROS reduction controls the cellular redox homeostasis, which plays a critical role in oxidative-stress-induced skeletal muscle atrophy [22,24,40]. Previous studies have shown that upregulation of Noxs protein expression participate in LPS-induced inflammatory responses and ROS production [40,41]. Nox4 is one of the major contributors of ROS generation in skeletal muscle [42]. The present study showed that LPS significantly enhanced Nox4 protein expression and decreased antioxidant SOD2 protein expression, leading to ROS overproduction in C2C12 myotubes, whose effects were inhibited by AN07 in the myotubes. Moreover, previous studies have revealed that AN07 is a PPARγ agonist [34] with antioxidant effects [39]. Consistently, our results also showed that GW9662, a PPARγ antagonist, but not AG1024 (IGF-1R inhibitor) or wortmannin (PI3K inhibitor), abrogated the antioxidant effects of AN07 in LPS-treated C2C12 myotubes. Taken together, these results indicate that AN07 effectively protects C2C12 myotubes against LPS-induced oxidative stress by reducing ROS accumulation and increasing antioxidant defense in a PPARγ-dependent and IGF-1/Akt-independent manner. In addition, inflamed muscles secrete several proinflammatory mediators. These molecules further induce oxidative stress or the protein degradation pathway, leading to muscle atrophy [13,14,26]. Furthermore, COX-2 is a critical enzyme responsible for the biosynthesis of PGE2, a well-known activator of inflammation. In this regard, attenuating the COX-2/PGE2 pathway serves as a potential pharmacological target against inflammation-induced muscle atrophy [43]. Our results showed that LPS significantly increased phosphorylation of IκBα and nuclear translocation of NF-κB and inflammatory mediators (TNF-α, IL-1β, COX-2, and PGE2) in LPS-induced atrophic myotubes. However, LPS-induced inflammatory signaling was attenuated by AN07. Similarly, the PPARγ antagonist GW9662, but not AG1024 or wortmannin, attenuated the inhibitory effect of AN07 on LPS-induced COX-2 expression in C2C12 myotubes. Collectively, the anti-inflammatory and antioxidant effects of AN07 against LPS-induced myotube atrophy were in a PPARγ-dependent manner.

In this study, we demonstrated that AN07 improved mitochondrial dysfunction in LPS-treated C2C12 myotubes. Mitochondrial respiration is the most important contributor of cellular energy. The dysfunction of mitochondria results in energy stress and is highly correlated with impairment of skeletal muscle mass and function [44]. Previous studies have demonstrated that oxidative stress induces mitochondrial dysfunction via the depolarization of inner mitochondrial membrane potential and subsequent impairment of oxidative phosphorylation [45]. In addition, LPS has been found to decrease the respiratory function in C2C12 myotubes [46,47]. In line with these reports, LPS not only increased oxidative stress but also decreased the mitochondrial membrane potential in C2C12 myotubes. In addition, LPS also decreased basal OCR, ATP production OCR, and maximal respiration OCR in C2C12 myotubes. However, AN07 significantly attenuated LPS-induced reduction of mitochondrial membrane potential and respiratory function, confirming the protective effects of AN07 on LPS-induced mitochondrial dysfunction.

IGF-1/IGF-1R/Akt signaling is a major protein synthesis pathway in skeletal muscle [7]. Phosphorylated Akt enhances protein synthesis of skeletal muscle by activating mTOR and its downstream signaling molecule S6 to stimulate translation initiation. The downregulation of IGF-1/IGF-1R/Akt signaling has been identified for its pathogenetic role in inflammation-induced muscle atrophy [8,10,21]. Our recent study showed LPS decreased IGF-1R/IRS-1/p-Akt expression but had no effects on mTOR and S6 phosphorylation [8]. This might be due to IGF-1 resistance after more than 18 h of LPS/IFN-γ treatment, while Akt was still sensitive to IGF-1 in myotubes [48,49]. In this study, LPS decreased IGF-1/IGF-1R/IRS-1/p-Akt signaling. We further measured p-mTOR/p-S6 after LPS treatment for 12 and 24 h. Results indicated neither p-mTOR nor p-S6 was affected by LPS treatment for 24 h, but we found both p-mTOR and p-S6 were downregulated by LPS treatment for 12 h. However, AN07 significantly attenuated LPS-induced myotube atrophy and upregulated IGF-1/IGF-1R/p-Akt/p-mTOR/p-S6 signaling. Moreover, the inhibitory effects and responsible mechanisms of AN07 on LPS-induced downregulation on MyHC expression and reduction of myotube diameter were attenuated by AG1024 or wortmannin, suggesting AN07 protects C2C12 myotubes against LPS via enhancing the IGF-1/IGF-1R/Akt pathway. Moreover, PPARγ has been recently found to be an upstream activator of IGF-1, and increased PPARγ can prevent angiotensin-II-mediated muscle atrophy [38]. In the present study, we further confirmed the involvement of PPARγ in the protein-synthesis-promoting effects of AN07 using GW9662. In line with the above findings, GW9662 attenuated the upregulatory effects of AN07 on the expression of IGF-1R and MyHC in LPS-treated C2C12 myotubes. Collectively, these results suggest the involvement of PPARγ and IGF-1R/Akt in the protein-synthesis-promoting effects of AN07 against LPS-induced atrophic effects.

In addition, protein degradation in the skeletal muscle is controlled by the activation of UPS and ALS pathway. Many studies have revealed that LPS induces skeletal muscle atrophy via the activation of the UPS pathway by enhancing the expression of atrogin-1 and MuRF1 [6,8,18,21]. Moreover, ALS is an evolutionary self-cleaning process that removes damaged organelles and protein aggregates to maintain healthy cells. LPS activates the ALS pathway, then leads to muscle atrophy [8,11,21,41]. Moreover, ALS and UPS are controlled by NF-κB and FoxO1a transcription factors, while the activity of FoxO1a is negatively regulated by Akt [6,13,50]. Given our observations that AN07 attenuated NF-κB-mediated inflammation while promoting Akt-mediated protein synthesis, we investigated the effect of AN07 on the LPS-induced protein degradation pathway. As expected, LPS increased the nuclear expression of both NF-κB and FoxO1a, leading to the upregulation of UPS- and ALS-related molecules, including atrogin-1, MuRF1, and LC3B-II/I. However, these effects were significantly attenuated by AN07 pretreatment. Furthermore, the addition of AG1024, wortmannin, and GW9662 abrogated the attenuating effect of AN07 on LPS-induced upregulation of MuRF1. These results suggest that AN07 protects C2C12 myotubes against LPS-induced atrophy via decreasing the protein degradation pathway, at least in part, in PPARγ- and IGF-1/Akt-dependent manners. In conclusion, AN07 is a potential antiatrophic agent for treating inflammation-induced skeletal muscle atrophy.

## 4. Materials and Methods

### 4.1. Materials

2-Hydroxy-4′-methoxychalcone (AN07) was kindly provided by Prof. Ferenc Fülöp (Institute of Pharmaceutical Chemistry, University of Szeged, Szeged, Hungary) and Prof. Fang-Rong Chang (Graduate Institute of Natural Products, Kaohsiung Medical University, Kaohsiung, Taiwan). The method for the synthesis of AN07 has been described in detail in previous studies [34,51]. The purity of AN07 was higher than 95% as determined by high-performance liquid chromatography. LPS (L8274) from *Escherichia coli* (O26:B6), dimethyl sulfoxide (DMSO), AG1024, GW9662, wortmannin, 2′,7′-dichlo-rodihydro-fluorescein diacetate (H_2_DCF-DA), and 3-(4, 5-dimethylthiazol-2-yl)-2,5-diphenyl-tetrazolium bromide (MTT) were obtained from Sigma-Aldrich Chemical Co. (St. Louis, MO, USA). Lactate dehydrogenase (LDH) cytotoxicity assay kit was purchased from G-Biosciences (St. Louis, MO, USA). 5,5′,6,6′-Tet-rachloro-1,1′,3,3′-tetraethyl benzimidazolyl carbocyanine iodide (JC-1), Dulbecco’s modified Eagle’s medium (DMEM), fetal bovine serum (FBS), horse serum (HS), all qRT-PCR reagents, T-PER tissue protein extraction reagent, and NE-PER nuclear and cytoplasmic extraction kit were purchased from Thermo Fisher Scientific (Waltham, MA, USA). The SDS-PAGE was performed using a Mini-PROTEAN^®^ electrophoresis system purchased from Bio-Rad (Hercules, CA, USA). Enhanced chemiluminescence (ECL) detection reagent and polyvinylidene difluoride (PVDF) membrane were acquired from Millipore (Billerica, MA, USA). Antibodies used in this study were as follows: β-actin (Sigma-Aldrich); MyHC (R&D system, Minneapolis, MN, USA); lamin B, Nox4, MuRF1, IκBα, COX-2, S6, p-S6 (Ser235/236), antigoat IgG-HRP, antirabbit IgG-HRP, and antimouse IgG-HRP (Santa Cruz, CA, USA); IGF-1R, IRS-1, p-IκBα, NFκBp65, p-Akt, Akt, p-mTOR (Ser2448), mTOR, LC3B, FoxO3a, SQSTM1/p62, SOD2, IL-1β, and TNF-α (Cell signaling, Danvers, MA, USA); atrogin-1 (ECM biosciences, Versailles, KY, USA); and Alexa Fluor 488 goat antimouse IgG secondary antibody (Thermo Fisher Scientific, Waltham, MA, USA).

### 4.2. Cell Culture and Drug Treatment

C2C12 myoblasts (CVCL_0188) were purchased from the Bioresource Collection and Research Center (Hsinchu, Taiwan). Cells were cultured in DMEM growth medium supplemented with 10% FBS, 100 U/mL penicillin, and 100 μg/mL streptomycin in a humidified incubator with 5% CO_2_ at 37 °C. At 80% of confluence, cells were cultured in differentiation media containing DMEM supplemented with 2% horse serum for 6 days to form myotubes. To investigate the protective effects of AN07 against LPS-induced damage, C2C12 myotubes were pretreated with vehicle (0.1% DMSO) or AN07 (0.01, 0.1, 0.5, and 1 μM) for 1 h and then treated with LPS (100 ng/mL) for 4, 12, or 24 h. For antagonist treatment, cells were pretreated with GW9662 (10 μM, dissolved in DMSO), AG1024 (10 nM, dissolved in DMSO), or wortmannin (100 nM, dissolved in DMSO) 1 h before AN07 treatment.

### 4.3. Cell Viability Assay

Cell viability was measured by MTT and LDH assay. The MTT assay depends on the reduction of MTT to form an insoluble purple formazan by metabolically active cells. Briefly, C2C12 myotubes were pretreated with AN07 (0.01–1 μM) for 24 h. Media were then collected for LDH assay, and the cells were treated with fresh medium containing MTT (final concentration: 0.5 mg/mL) at 37 °C for 2 h. The formazan crystals were dissolved with DMSO, and the absorbance was measured at 540 nm using an ELISA reader (Multiskan Ascent microplate reader, Thermo Fisher Scientific, Waltham, MA, USA). In LDH assay, culture medium was collected to measure LDH levels using a cytotoxicity detection kit. The tetrazolium salts produced in LDH-induced enzymatic reaction were then reduced to red formazan, thereby allowing colorimetric detection by the ELISA reader at 490 nm.

### 4.4. Quantitative RT-PCR (qRT-PCR)

TRIzol reagent was used to extract total RNA, and 1 μg RNA was reverse-transcribed into cDNA using a high-capacity cDNA reverse transcription kit. qRT-PCR was performed by StepOnePlus™ real-time PCR system (Thermo Fisher Scientific, Waltham, MA, USA) with fast SYBR green master mix. Melting curve analysis was performed to monitor PCR product purity. The mRNA levels of IGF-1 were normalized to the level of the housekeeping gene glyceraldehyde-3-phosphate dehydrogenase (Gapdh). Specific primers and their sequences were as follows: Gapdh forward 5′-TGTCAAG-CTCATTTCCTGGT-3′, reverse 5′-TAGGGCCTCTCTTGCTCAGT-3′; Igf-1 forward 5′-GGACCGAGGGGCTTTTACTT-3′, reverse 5′- TCCGGAAGCAACACTCATCC -3′.

### 4.5. Measurement of Intracellular Reactive Oxygen Species (ROS) and Mitochondrial Membrane Potential (ΔΨm)

C2C12 myotubes were cultured in 96-well plates and pretreated with AN07 (0.01–1 μM) for 1 h and then stimulated with LPS (100 ng/mL) for 24 h. H_2_DCF-DA fluorescence staining was used to measure intracellular ROS production, and DCF fluorescence signals were detected by a BioTek Synergy H1 microplate reader (BioTek, Winooski, VT, USA) at 495/520 nm (excitation/emission). DCF fluorescence are represented as percentages of the vehicle group. The ΔΨm was measured using the fluorescent cationic dye JC-1. The changes in JC-1 fluoresce ratio between red (540/570 nm) and green (495/520 nm) were measured by a BioTek Synergy H1 microplate reader and normalized to the vehicle group.

### 4.6. Western Blot Analysis

Cells were lysed in T-PER tissue protein extraction buffer to obtain total cell extract or NE-PER nuclear and cytoplasmic extraction kit to isolate nuclear fraction. Protein concentration was measured by Bradford protein assay. Then, proteins were fractionated by SDS-PAGE, and the separated proteins were transferred onto PVDF membranes. Membranes were blocked in 5% nonfat milk in TBST buffer (50 mM Tris-HCl, pH 7.6, 150 mM NaCl, 0.1% Tween 20) with 1 h shaking at room temperature. The blot was probed with primary antibodies (diluted 1:1000 in 2% BSA) overnight at 4 °C. Subsequently, membranes were washed with TBST buffer and hybridized with secondary HRP-conjugated antibody (diluted 1:10000 in 5% nonfat milk) with shaking at room temperature for 1 h. Protein expression was detected with chemiluminescence using an Amersham ECL detection kit, and the signal was captured using the TopBio Multigel-21 chemiluminescent imaging system (Topbio, Taiwan) and quantified with the ImageJ software (National Institutes of Health, Bethesda, MD).

### 4.7. Measurement of PGE2 Production

PGE2 production in culture medium supernatant was measured by a PGE2 EIA kit following the manufacturer’s guide. Briefly, C2C12 myotubes were pretreated with AN07 (0.01–1 μM) for 1 h and then stimulated with LPS (100 ng/mL) for 24 h. Media were then collected, and the absorbance at 405 nm was measured using an ELISA reader.

### 4.8. Immunofluorescence Staining

The morphology of matured C2C12 myotubes were observed by immunofluorescence staining. After fixing in 4% paraformaldehyde in PBS (pH 7.4) for 10 min at room temperature, the myotubes were permeabilized by 0.2% Triton X-100 in PBS for 15 min at room temperature. To decrease nonspecific binding, myotubes were blocked with 3% BSA in PBS for 1 h at room temperature and then hybridized with mouse anti-MyHC antibody (1:1000 in 0.1% BSA) overnight at 4 °C. After that, Alexa Fluor 488 conjugated secondary antibody (1:1000 in 0.1% BSA) was added for 1 h at room temperature. Images were captured using a Leica DMi8 inverted light microscope (Leica DFC 7000T sCMOS, Wetzlar, Germany) at 100x magnification controlled by Leica Application Suite X software (Version 3.0.3). For each experimental condition, five random fields were chosen, and at least 60 myotubes were analyzed for the diameters using Image J software.

### 4.9. Mitochondrial Bioenergetics

Seahorse XFp analyzer (XFp, Seahorse Biosciences, MA) was used to evaluate the mitochondria oxygen consumption rate (OCR) of C2C12 myotubes. After C2C12 myoblasts were seeded and differentiated in Seahorse XFp miniplates, myotubes were pretreated with AN07 (0.1 or 1 μM) for 1 h followed by the addition of LPS (100 ng/mL) for 24 h. Before assay, the fluorescence probes were activated by Agilent Seahorse XF calibrant at 37 °C in a non-CO2 incubator overnight. Subsequently, the culture medium was replaced with Agilent Seahorse XF base medium (with 1 mM pyruvate, 2 mM glutamine, 10 mM glucose, and pH adjusted to 7.4), and the miniplates were then placed in a non-CO_2_ incubator at 37 °C for 1 h. The key parameters of mitochondrial function were measured using Seahorse XFp Cell Mito Stress Test, which were obtained at the baseline and following sequential injection of oligomycin (1 μM), FCCP (1 μM), and a mixture of antimycin A plus rotenone (AA/ROT, 1 μM).

### 4.10. Statistical Analysis

The results are expressed as mean ± SEM from six independent observations, and the statistical significance was analyzed by one-way ANOVA with post-hoc Tukey HSD test or Kruskal–Wallis test with Dunn’s test as indicated. All the statistical analyses were performed using InStat version 3.0 (GraphPad Software, San Diego, CA, USA). Significant difference was set at *p* < 0.05. The data of MTT assay, Western blotting, and qRT-PCR were all normalized by changing into “percentage matched control values“ to reduce unwanted sources of variation.

## 5. Conclusions

This study demonstrated that AN07 attenuated LPS-induced reduction of myotube diameter, oxidative stress, mitochondrial dysfunction, inflammatory response, and protein catabolism in C2C12 myotubes. AN07 improved the mitochondrial membrane potential and mitochondrial oxygen consumption rate. AN07 decreased the production of ROS and inflammatory mediators while enhancing the antioxidant defense. Furthermore, AN07 enhanced protein synthesis and inhibited protein degradation via increasing the IGF-1/IGF-1R/Akt-mediated protein synthesis pathway and decreasing the UPS- and ALS-related protein degradation pathways. These findings suggest that AN07 might be a potential agent for improving inflammation-induced skeletal muscle atrophy.

## Figures and Tables

**Figure 1 ijms-23-12929-f001:**
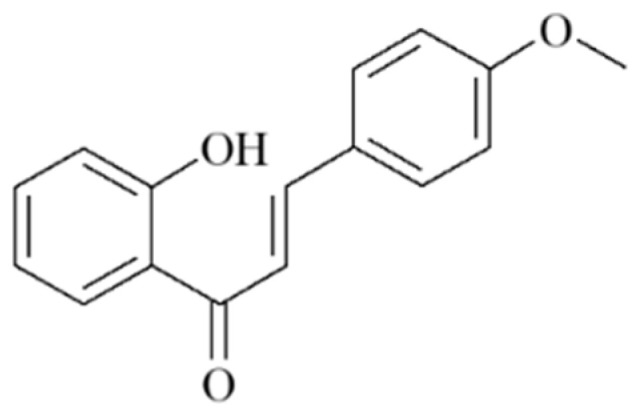
Chemical structure of 2-hydroxy-4’-methoxychalcone (AN07).

**Figure 2 ijms-23-12929-f002:**
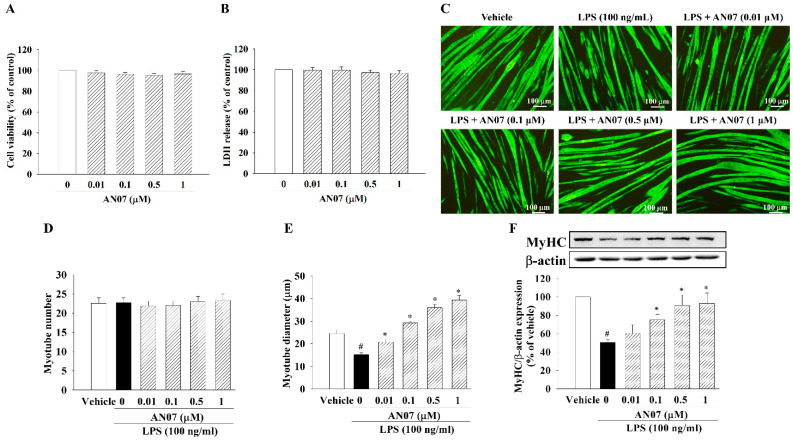
AN07 reduced LPS-induced C2C12 myotube atrophy. Myotubes were treated with AN07 (0.01–1 μM) for 24 h. Effects of AN07 on (**A**) cell viability and (**B**) cytotoxicity of C2C12 myotubes were determined by MTT assay and LDH assay, respectively. (**C**) Immunofluorescence staining of MyHC (green) was performed to visualize C2C12 myotube morphology. Myotubes were pretreated with AN07 (0.01–1 μM) for 1 h followed by LPS (100 ng/mL) treatment for 24 h. (**D**) Myotube number and (**E**) diameter were quantified. Scale bar = 100 μm. (**F**) MyHC protein expression was detected by Western blotting. Densitometry analyses are presented as the relative ratio of protein/β-actin protein and are represented as percentages of the control group. Data represent the mean ± SEM from six independent experiments (n = 6). ^#^
*p* < 0.05 vs. control group (vehicle only); * *p* < 0.05 vs. LPS-treated group according to one-way ANOVA followed by a Tukey post-hoc test (**D**,**E**) or Kruskal–Wallis test followed by Dunn’s test (**A**,**B**,**F**).

**Figure 3 ijms-23-12929-f003:**
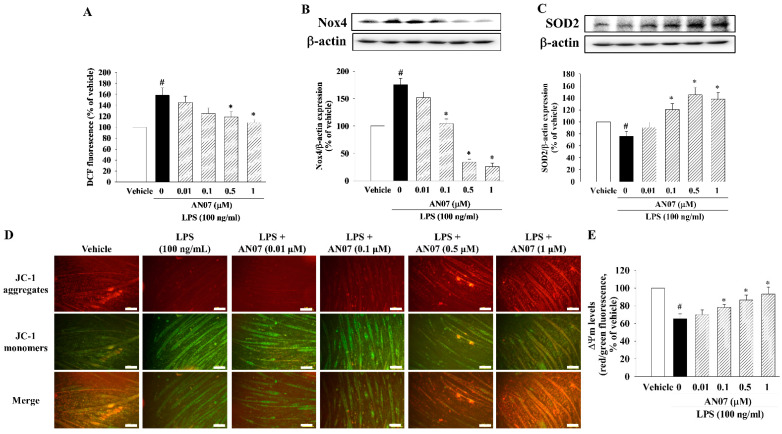
AN07 attenuated LPS-induced oxidative stress in C2C12 myotubes. Myotubes were pretreated with AN07 (0.01–1 μM) for 1 h followed by LPS (100 ng/mL) treatment for 24 h. (**A**) ROS levels were measured by DCF fluorescence using a microplate reader at 495/520 nm (excitation/emission). Protein expressions of (**B**) Nox4 and (**C**) SOD2 were measured by Western blotting. Densitometry analyses are presented as the relative ratio of protein/β-actin protein and are represented as percentages of the control group. (**D**,**E**) Mitochondrial membrane potential levels were detected by JC-1 staining. (**D**) Representative images of fluorescence of JC-1 in C2C12 myotubes. Scale bar = 100 μm. (**E**) Quantification of the ratio of JC-1 red/green fluorescence intensity using a microplate reader at red (540/570 nm) and green (495/520 nm) in C2C12 myotubes. Data represent the mean ± SEM from six independent experiments (n = 6). ^#^
*p* < 0.05 vs. control group (vehicle only); * *p* < 0.05 vs. LPS-treated group according to Kruskal–Wallis test followed by Dunn’s test.

**Figure 4 ijms-23-12929-f004:**
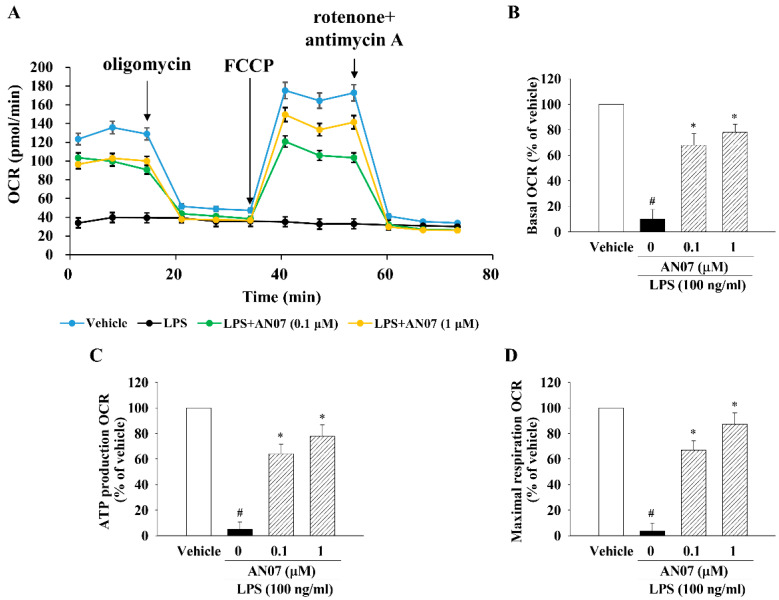
AN07 improved LPS-induced reduction of mitochondrial oxygen consumption rate (OCR) in C2C12 myotubes. Myotubes were pretreated with AN07 (0.1 and 1 μM) for 1 h followed by LPS (100 ng/mL) treatment for 24 h. OCR was measured by Seahorse XFp analyzer. Oligomycin (1 μM), FCCP (1 μM), and a rotenone/antimycin-A mix (1 μM) were added at times indicated by arrows. (**A**) Representative readouts of OCR. Quantification of (**B**) basal OCR, (**C**) ATP production OCR, and (**D**) maximal respiration OCR. Data represent the mean ± SEM from six independent experiments (n = 6). ^#^
*p* < 0.05 vs. control group (vehicle only); * *p* < 0.05 vs. LPS-treated group according to Kruskal–Wallis test followed by Dunn’s test.

**Figure 5 ijms-23-12929-f005:**
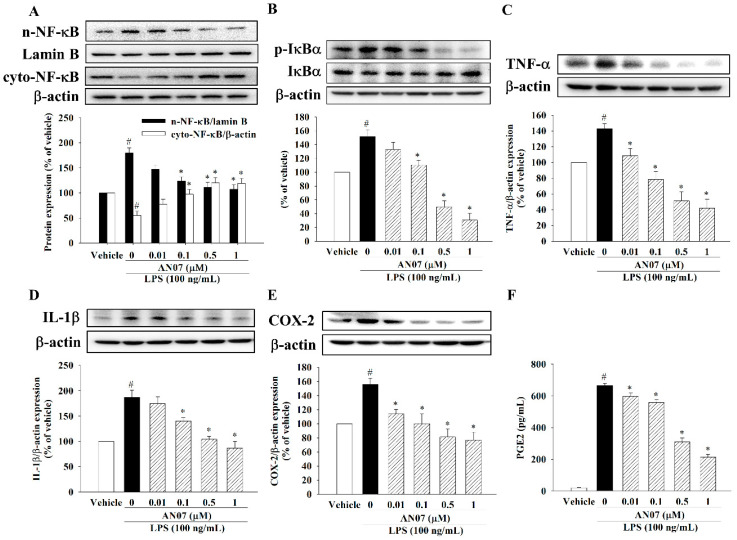
AN07 attenuated LPS-induced inflammatory signaling in C2C12 myotubes. Myotubes were pretreated with AN07 (0.01–1 μM) for 1 h followed by LPS (100 ng/mL) treatment for 2 h. The protein expressions of (**A**) nuclear (n-) and cytosolic (cyto-) NFκBp65, (**B**) p-IκBα/IκBα, (**C**) TNF-α, (**D**) IL-1β, and (**E**) COX-2 were measured by Western blotting. Densitometry analyses are presented as the relative ratio of protein/β-actin or lamin B and are represented as percentages of the control group. (**F**) For measuring PGE2 levels, myotubes were pretreated with AN07 (0.01–1 μM) for 1 h followed by LPS (100 ng/mL) treatment for 24 h. Then, the culture media were collected for measuring PGE2 levels by ELISA assay. Data represent the mean ± SEM from six independent experiments (n = 6). ^#^
*p* < 0.05 vs. control group (vehicle only); * *p* < 0.05 vs. LPS-treated group according to Kruskal–Wallis test followed by Dunn’s test (**A**–**E**) or one-way ANOVA followed by a Tukey post-hoc test (**F**).

**Figure 6 ijms-23-12929-f006:**
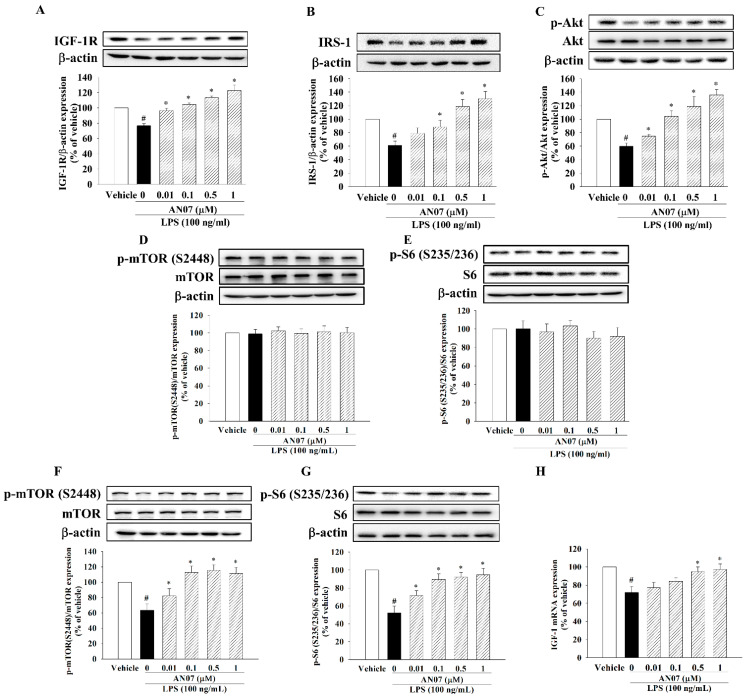
AN07 improved LPS-induced reduction of protein synthesis pathway in C2C12 myotubes. Myotubes were pretreated with AN07 (0.01–1 μM) for 1 h followed by LPS (100 ng/mL) treatment for 24 h. The protein expressions of (**A**) IGF-1R, (**B**) IRS-1, (**C**) p-Akt/Akt, (**D**) p-mTOR (S2448)/mTOR, and (**E**) p-S6 (S235/236)/S6 were measured by Western blotting. (**H**) The IGF-1 mRNA expression was measured by qPCR. (**F**,**G**) For short-term LPS treatment, myotubes were pretreated with AN07 (0.01–1 μM) for 1 h followed by LPS (100 ng/mL) treatment for 12 h. The protein expressions of (**F**) p-mTOR (S2448)/mTOR and (**G**) p-S6 (S235/236)/S6 were measured by Western blotting. Densitometry analyses are presented as the relative ratio of protein/β-actin or p-protein/protein and are represented as percentages of the control group. Data represent the mean ± SEM from six independent experiments (n = 6). ^#^
*p* < 0.05 vs. control group (vehicle only); * *p* < 0.05 vs. LPS-treated group according to Kruskal–Wallis test followed by Dunn’s test.

**Figure 7 ijms-23-12929-f007:**
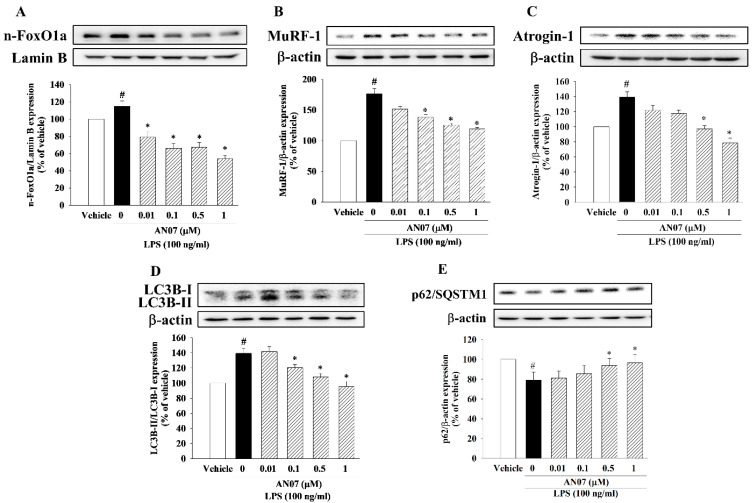
AN07 attenuated LPS-induced protein degradation pathway in C2C12 myotubes. Myotubes were pretreated with AN07 (0.01–1 μM) for 1 h followed by LPS (100 ng/mL) treatment for 24 h. The protein expressions of (**A**) nuclear FoxO1a, (**B**) MuRF1, (**C**) atrogin-1, (**D**) LC3B-II/LC3B-I, and (**E**) p62/SQSTM1 were measured by Western blotting. Densitometry analyses are presented as the relative ratio of protein/β-actin or lamin B and are represented as percentages of the control group. Data represent the mean ± SEM from six independent experiments (n = 6). ^#^
*p* < 0.05 vs. control group (vehicle only); * *p* < 0.05 vs. LPS-treated group according to Kruskal–Wallis test followed by Dunn’s test.

**Figure 8 ijms-23-12929-f008:**
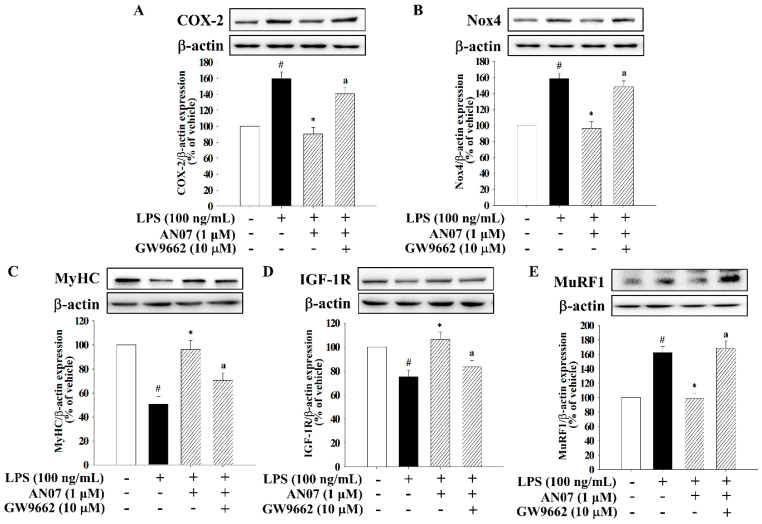
GW9662 *(PPARγ* antagonist) attenuated the anti-inflammatory, antioxidant, and antiatrophic effects of AN07 in LPS-treated C2C12 myotubes. Myotubes were treated with or without GW9662 (10 μM) 1 h before AN07 (1 μM) treatment for 1 h and then treated with LPS (100 ng/mL) for (**A**) 2 h or (**B**–**E**) 24 h. The protein expressions of (**A**) COX-2, (**B**) Nox4, (**C**) MyHC, (**D**) IGF-1R, and (**E**) MuRF1 were measured by Western blotting. Densitometry analyses are presented as the relative ratio of protein/β-actin and are represented as percentages of the control group. Data represent the mean ± SEM from six independent experiments (n = 6). ^#^
*p* < 0.05 vs. control group (vehicle only); * *p* < 0.05 vs. LPS-treated group; ^a^
*p* < 0.05 vs. LPS+AN07-treated group according to Kruskal–Wallis test followed by Dunn’s test.

**Figure 9 ijms-23-12929-f009:**
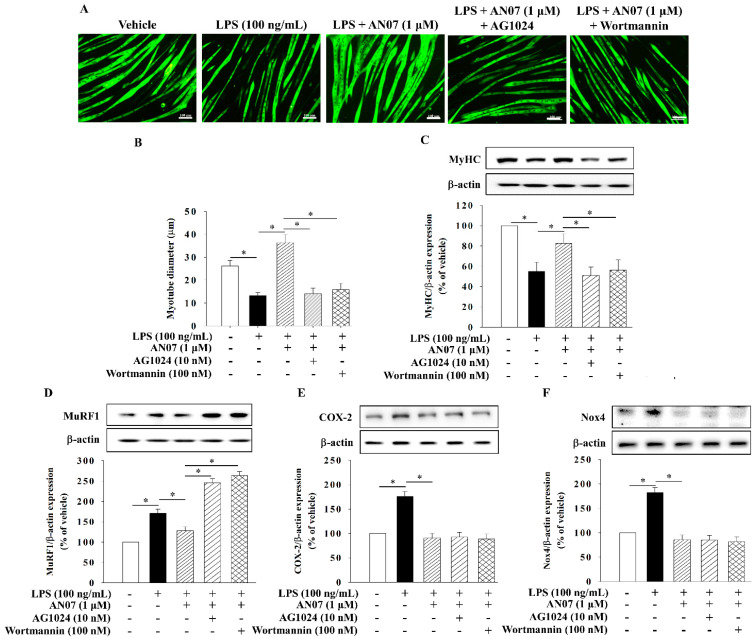
AG1024 (IGF-1R inhibitor) and wortmannin (PI3K inhibitor) attenuated the AN07-mediated antiatrophic effects and MuRF1 downregulation in LPS-induced C2C12 myotubes. Myotubes were treated with or without AG1024 (10 nM) or wortmannin (100 nM) for 1 h before AN07 (1 μM) treatment for 1 h and then treated with LPS (100 ng/mL) for (**E**) 2 h or (**A**–**D**,**F**) 24 h. (**A**) The morphology of myotube were visualized by immunofluorescence staining of MyHC (green). (**B**) Myotube diameter were quantified. Scale bar = 100 μm. The protein expressions of (**C**) MyHC, (**D**) MuRF1, (**E**) COX-2, and (**F**) Nox4 were measured by Western blotting. Densitometry analyses are presented as the relative ratio of protein/β-actin and are represented as percentages of the control group. Data represent the mean ± SEM from six independent experiments (n = 6). * *p* < 0.05 indicated significant difference between groups according to one-way ANOVA followed by a Tukey post-hoc test (**B**) or Kruskal–Wallis test followed by Dunn’s test (**C**–**F**).

## Data Availability

The data presented in this study are available on request from the corresponding author.

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
