# Peer review of "Protective Effects of the Chalcone-Based Derivative AN07 on Inflammation-Associated Myotube Atrophy Induced by Lipopolysaccharide"

_ijms, 2022, doi:10.3390/ijms232112929_

Round 1

Reviewer 1 Report

Introduction: 90 % of this section is about inflammation, ROS, and other related enzymes, but very little is about the chalcone, improve and make a logical relationship. 

In the introduction, there is no discussion about the advantages and disadvantages of the recently reported methods. What is the problem of the previous studies that need to be solved? What is the meaning and innovation of this work? The advance of this work compared with other works should be given in detail.

Reviewer 2 Report

The current manuscript described the AN07 which is a synthetic chalcone and reported for anti-inflammatory and neuroprotective in the previous literature.  Herin authors explored its anti-inflammatory properties for treating myotube atrophy. I recommend it with minor suggestions.

1. This sentence in abstract is not making proper sense "However, both IGF-1R inhibitor AG1024 and PI3K inhibitor wortmannin inhibited AN07-induced protein synthesis/degradation signaling. Finally, PPARy antagonist GW9662 inhibited AN07 protection on LPS-induced inflammation, oxidation and protein catabolism."

2. What about if authors study the COX-1 enzyme inhibition?

3. "Cox-2" can be correct as "COX-2" throughout the manuscript.

Reviewer 3 Report

1. In Figure 4, why are only 2 concentrations of AN-07 included, not 4 concentrations like in Figure 2,3,5,6.

2. In Figure 5A, quantification of cyto-NF-kB/beta-actin should be added.

Round 2

Reviewer 1 Report

The content of paper was well organized, all the suggested points are incorporated and easy for the reader to follow the subject discussed, thus support for its acceptance.